# Interception and Redistribution of Precipitation by *Parkinsonia aculeata* L.: Implications for Palo Verde National Park Wetlands, Costa Rica

**Julio César Calvo-Alvarado** [1,2], **César Dionisio Jiménez-Rodríguez** [3,4,*], **Juan Carlos Solano** [1] and **Oscar Arias-Rodríguez** [2]

1   Escuela de Ingeniería Forestal, Tecnológico de Costa Rica, Cartago 159-7050, Costa Rica; jucalvo@tec.ac.cr (J.C.C.-A.); solano.juancarlos@gmail.com (J.C.S.)
2   Organization for Tropical Studies, San José 676-2050, Costa Rica; oariasro@gmail.com
3   Environmental Research and Innovation (ERIN) Department, Luxembourg Institute of Science and Technology, L-4422 Luxembourg, Luxembourg
4   Water Resources Section, Delft University of Technology, 2628 Delft, The Netherlands
*   Correspondence: cdjimenezcr@gmail.com

**Abstract:** Seasonal wetlands in the tropics are important habitats for local and migratory bird species. In the northwestern Pacific of Costa Rica, Palo Verde National Park has one of the most important seasonal wetlands of Central America. The management history of this wetland has shown the impact of invasive plant species such as *Parkinsonia aculeata* L. whose cover extension and canopy structure impact not only the ecological niches of bird species, but also the wetland hydrology. A 300 m$^2$ plot was established in a *P. aculeata* stand to evaluate the role of *P. aculeata* on the partitioning and redistribution of precipitation. Gross precipitation ($P_{Gr}$), throughfall ($P_{TF}$) and stemflow ($P_{SF}$) were measured on a daily basis to determine the interception of precipitation ($P_I$) and net precipitation ($P_{Net}$). A total of 43 precipitation events were sampled during the wet season of 2003. We measured 530.5 mm of $P_{Gr}$ and 458 mm of $P_{TF}$, with an average sampling error of 0.7 mm or 6.1%. Canopy storage capacity was estimated at 1.47 mm, throughfall 88.73%, stem flow 2.63% and a total interception of 8.64%, with a $P_{Net}$ coefficient of 0.9475. The relationships between gross precipitation ($P_{Gr}$) with throughfall ($P_{TF}$), stemflow ($P_{SF}$) and net precipitation ($P_{Net}$) were evaluated using linear regression models. *P. aculeata* showed to have one of the highest net precipitation and lowest precipitation interception among small trees.

**Keywords:** interception; canopy; dry forest; seasonal wetland; Central America

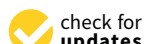



## 1. Introduction

Palo Verde National Park (PVNP) protects an area of 24,519 ha which includes one of the largest and most important seasonal wetland systems of Central America [1]. The wetland system is composed of 13 water bodies for a total area of ~10,000 ha. This wetland system is home to more than 60 species of native and migratory birds and provides numerous ecosystemic goods and services. The conservation and restoration of these wetlands to maintain healthy populations of migratory birds and other species has become a critical component in the PVNP management goals. Because of this, the PVNP was included as a RAMSAR site in 1991 and in the Montreux list of Important International Wetlands in 1993 [2–4].

The PVNP was originally a cattle ranch with livestock larger than 10,000 heads before it became a national park in 1970s. Thereafter, active management activities such as cattle grazing and the use of fire were forbidden when the national park was declared [5–8]. These major changes, plus a sequence of droughts, modified the hydrological dynamic of the wetlands, leading to the propagation of invasive plants such as *Typha domingensis*

Pers. and *Parkinsonia aculeata* L. [3,9,10]. In terms of vegetation cover change, the Palo Verde Lagoon, a non-tidal marsh, was almost free of *T. domingensis* and *P. aculeata* before the 1980s [6,9]. However, after more than 40 years without active management of the wetland, a large portion of the lagoon is covered by *T. domingensis* while *P. aculeata* invaded most of the shallow littoral zone together with the woody shrub *Mimosa pigra* L. [5–8,11–13]. The invasion of these plant species has changed the wetland vegetation cover, reducing the original areas of feeding, shelter and nesting areas for migratory and native aquatic bird species. According to Solano [13], by the year 2003 there was about 356 ha of *P. aculeata* forest equivalent to 28% of the total area of this wetland. This forest had a tree density of $1338 \pm 38$ trees ha$^{-1}$, with a mean tree height of 3.5 m (range 1.8–5.8 m) and a tree diameter of 5.37 cm (range 2.4–15 cm).

The seasonal arrival of migratory and native aquatic birds takes place just when the flooded area of the wetland reaches its maximum water level at the end of the rainy season [14]. In this way the shoreline retreat process extends for a longer time during the dry season. Consequently, any water losses from this wetland during the rainy or dry seasons, driven by invasive plants such as *T. domingensis* and *P. aculeata*, also must be understood. During the last decades several hydrological studies have been carried out in PVNP. A preliminary water balance of the Palo Verde Lagoon was carried out by Guzman [15], Calvo Alvarado and Arias [16], who studied the evapotranspiration of *T. domingensis*, while Jiménez-Rodríguez et al. [17] addressed the evaporation rates of different macrophytes present on the lagoon. However, to the best knowledge of the authors, no study has been carried out evaluating the rainfall interception capacity of *P. aculeata* either worldwide or in Costa Rica. All this points to that the potential negative impact from rainfall interception of *P. aculeata* is unknown and it could be considerable in the water balance of Palo Verde Lagoon or any other wetland.

*P. aculeata* is a small tree that can be either single-stemmed or multi-stemmed, usually with 5 m to 7 m of maximum height and typically is multi-trunked with a brown, fissured, or scaly bark with age. The crown is broad and showy with hanging branches when aged. The bark of young branches, twigs and younger stems is thin, smooth, and yellow-green or blue-green. The leaves are shortly stalked and bipinnate, each with one to four pairs of pinnae stiff, needle-sharp spines (modified leaf rachises) of 5–15 mm long formed at the base of each leaf, persisting on the older branches and stems. The species is originally from America but after 1700s this tree was introduced around the world as an ornamental, hedging, fodder and shade tree. Now it is a pan-tropical tree species considered invasive due to its extraordinary ability to survive and grow under a wide range of environmental conditions [18–20].

*P. aculeata* is considered as a water stress tolerant species [21], it tends to form dense, thorny, impenetrable thickets along drainage lines, depressions, and seasonal wetlands [22]. *P. aculeata* can continue to grow with their root systems inundated by water and can survive more than nine months with the lower portion of their trunks under water. In addition, the propagation of this species is remarkable. Seed production begins when plants are about 1.5 m tall and increases greatly with plant size at any one site and year. Large trees can produce at least 2500 seeds m$^{-2}$ of projected canopy cover. Pods fall close to the parent plant and can float for up to 14 days. Finally, this species can reshoot laterally from the main stem after damage such as after fire [18–20].

Undoubtedly, water is the driving factor of seasonal wetlands [23–25] such as the Palo Verde Lagoon Wetland of PVNP. Therefore, and given the extent of *P. aculeata* in these wetlands, quantifying the redistribution of precipitation by interception of this invasive tree species is critical for understanding the water balance dynamics of this wetland. Hence, the aim of this work is (i) to evaluate the precipitation interception fluxes of *P. aculeata*, (ii) to study the effect of forest structure in the partitioning of precipitation, and (iii) to develop linear regression equations to estimate these water fluxes useful for modeling the impact of *P. aculeata* on the water balance of Palo Verde Lagoon.

## 2. Materials and Methods

Our study focuses on the *P. aculeata* forest stands located on the littoral zone of Palo Verde Lagoon (10°20′35″ N, 85°20′25″ W). This wetland is important due to its size and very flat uniform topography that maximizes the wetland open water area at the end of the rainy season. The deepest section is at 8.1 m a.s.l. and the maximum elevation quota for the flooded area is at 11 m a.s.l., being the Tempisque levees above 12 m a.s.l. [26]. The soils are Vertisols [2], very fertile with expanded clays and minimum infiltration capacity [15,27,28]. The PVNP is classified as Tropical Dry Forest according to the Holdridge Life Zone System [29], with a well-defined dry season of five to six months (December to April). Data from the OTS automatic weather station indicates for the period 2000–2005 a mean annual precipitation of 1023.6 ± 230.8 mm yr$^{-1}$ and an average annual temperature of 27.6 ± 0.3 °C, with the year 2003 registering 1400.4 mm yr$^{-1}$ of rain.

### 2.1. Experimental Design

A fragment of *P. aculeata* forest was selected near to the Organization for Tropical Studies (OTS) research station in PVNP (10°20′37″ N, 85°20′11″ W), at an elevation of 9 m.a.s.l. A previous forest inventory of *P. aculeata* stands in the Palo Verde Lagoon concluded that a 10 m × 10 m plot is statistically accurate to describe forest structure characteristics of pure stands [13]. Hence, in the selected fragment of *P. aculeata* stand, a temporal plot of 300 m$^2$ (10 m × 30 m) was established to carry out the trial. The plot was divided in three subplots of 100 m$^2$ (10 m × 10 m) to study the effect of forest structure (Figure 1). The subplot forest structure was characterized by tree height (m) and tree diameter (cm) at 0.3 m above the base of the tree to avoid dealing with multiple stems above that height. Thereafter, we estimated the tree density (trees ha$^{-1}$), basal area (m$^2$ ha$^{-1}$), mean plot tree diameter (cm) and mean plot tree height (m). In each subplot 10 plastic funnel-type gauges were randomly distributed and placed at 1.5 m above the ground to measure throughfall ($P_{TF}$). The height at which the gauges were placed on site was chosen with the aim of continuously measuring even during the flooded periods of the lagoon shore.

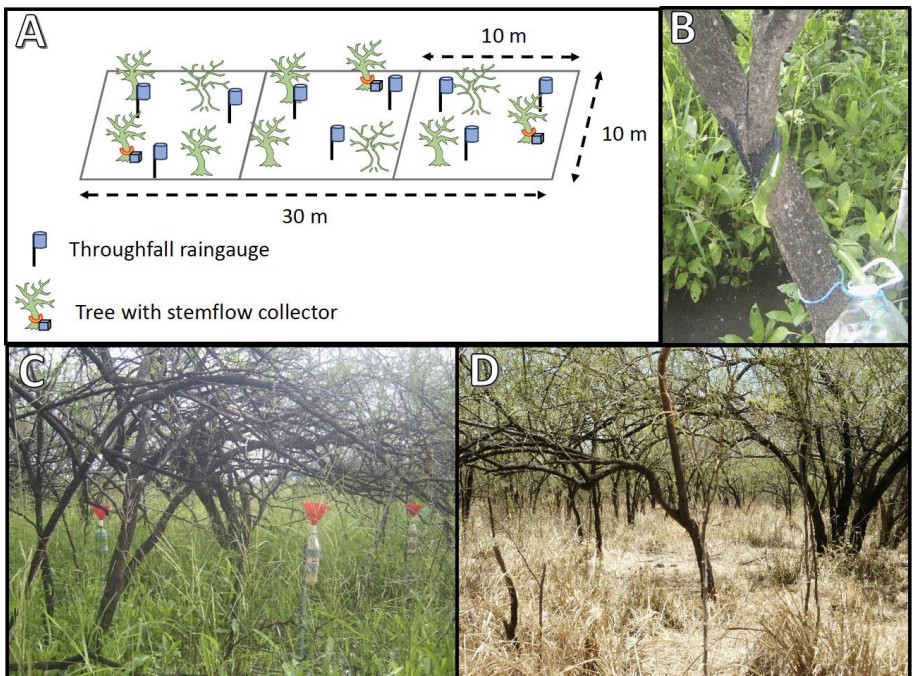

**Figure 1.** Details of the experimental setup used to investigate the interception and redistribution of precipitation in *Parkinsonia aculeata* stands at Palo Verde National Park, Costa Rica. (**A**) shows the plot design, (**B**) depicts the collar used to collect the stemflow, (**C**) shows the elevated rain gauges used to collect throughfall, and (**D**) provides a glimpse of the same plot during dry season.

A revolving sampling technique was implemented to reduce the standard error of the estimation caused by the variability in the horizontal distribution of tree crowns. Hence, all throughfall gauges were redistributed randomly three times during the trial [30,31]. Gross precipitation ($P_{Gr}$) was measured by placing one plastic funnel rain gauge in the nearest forest gap at ~100 m of distance. The volume of water collected by each throughfall rain gauge was converted into mm d$^{-1}$ with respect to the funnel collection area. The throughfall gauges were built using a plastic funnel of 16.5 cm of diameter attached to a plastic container of 2.0 L.

Stemflow ($P_{SF}$) was evaluated by selecting five single stem trees representative of the tree diameter distribution within the 300 m$^2$ plot. The projected crown area of the selected trees was estimated with the average of five projected crown diameters. The stemflow was collected through a spiral plastic collar attached to the trunks with nails and sealed with silicone paste. Each spiral collar conducted the stemflow to a plastic container of 20 L placed at the foot of the tree. The collected volume of water in L per tree was converted into mm d$^{-1}$ with respect to estimated tree crown area. Stemflow was projected to an area of 1 m$^2$ by calculating the average stemflow (mm) of all evaluated trees and multiplying it by the mean number of trees per square meter per plot. The sampling period of this study was from August to November 2003 and all daily measurements were conducted between 6:00 a.m. to 7:00 a.m. The rainfall events that finished after or before the measuring time were assigned to the precipitation of the previous day. Consequently, this manuscript considers the daily precipitation (mm d$^{-1}$) as an event and not multiple individual events within the same day.

*2.2. Data Analysis*

Net precipitation ($P_{Net}$) was defined as the water that reached the forest floor and it corresponds to the sum of throughfall ($P_{TF}$) and stemflow ($P_{SF}$) (Equation (1)). The relationship among gross precipitation ($P_{Gr}$) and all water fluxes was evaluated through linear regression models. Linear regression models applied for $P_{TF}$ and $P_{Net}$ following the equation were defined by $y_i = \alpha + \beta x_i + \epsilon_i$, where $\alpha$ is the interception with the y axis, $\beta$ is the slope of the equation that represents the proportion of water retained by the canopy, and $\epsilon_i$ is the error associated with the estimate. Regardless of if the throughfall and $P_{Net}$ equation's intercepts were non-significant ($p = 0.05$) different from cero, we decided to keep the intercept in the regression model to avoid altering the true meaning of the slope. In addition, a polynomial regression model was applied to $P_{SF}$ as $y_i = \alpha + \beta x_i + \gamma x_i^2 + \epsilon_i$. In this case, the $\alpha$ term was forced to zero considering that with no precipitation there is no possibility for the forest stand to yield stemflow. In all the regression models applied, the terms $\alpha$, $\beta$, and $\gamma$ were tested to determine if they were statistically different from zero ($p < 0.05$). We also compared the mean values of $P_{TF}$ among subplots with an analysis of covariance (ANCOVA) applying a Tukey Post Hoc test to determine the statistical different among subplots ($p = 0.05$). Canopy storage capacity ($S_c$) was calculated as the interception with the x axis of Equation (2) according to Loescher et al. [32]. We calculated the sampling error ($\epsilon_i$) in percentage (%) according to Jiménez-Rodríguez and Calvo [33] to evaluate the precision of the sampling design. This estimation of error is based on the daily throughfall average of the whole plot with respect to daily gross precipitation.

$$P_{Net} = P_{TF} + P_{SF}, \tag{1}$$

$$\log_{10} P_{Net} = \varphi + \omega \log_{10} P_{Gr}, \tag{2}$$

## 3. Results and Discussion

The whole plot and subplots fell between the ranges of the 2003 forest inventory (total of 40 plots) for tree density, diameter, basal area, and height. Except for tree diameter, all average values of our study are above the estimated values in the survey of Solano [13]. For tree density, the subplot 3 resulted in values of 4000 trees ha$^{-1}$, well above the maximum of 3500 trees ha$^{-1}$ of the 2003 forest inventory (Table 1).

**Table 1.** Structural characteristics for the subplot and whole plot of *Parkinsonia aculeata* L. in Palo Verde National Park, Costa Rica.

| Subplot | Trees per Plot | Tree Density (tree ha$^{-1}$) | Basal Area (m$^2$ ha$^{-1}$) | Tree Diameter (cm) | Tree Height (m) |
|---|---|---|---|---|---|
| 1 | 9 | 900 | 9.09 | 11.34 | 4.6 |
| 2 | 32 | 3200 | 4.66 | 4.31 | 3.95 |
| 3 | 40 | 4000 | 6.44 | 4.53 | 3.84 |
| Whole Plot | 27 | 2700 | 9.59 | 6.73 | 4.13 |
| 2003 Forest Survey [13] | 16 | 1600 | 8.42 | 8.12 | 3.88 |

The trial ran from the end of August to the end of November 2003 (115 days), sampling a total of 46 days with precipitation within that period. The gross precipitation gauge collected 627.9 mm with a daily maximum of 80.1 mm d$^{-1}$ and a daily minimum of 0.61 mm d$^{-1}$. According to the automatic OTS's Palo Verde weather station for the period 2007–2020, 83% of daily precipitations were under 5 mm d$^{-1}$ and 99% were under 90 mm d$^{-1}$. Thus, the sampling conducted in this study represents very well the long-term daily precipitation distribution of the site.

Unfortunately, during the trial we lost the three largest precipitation events that overflowed all stemflow gauges (Figure 2A). Consequently, for the sake of a balanced and comparable analysis of the water fluxes balance and the regression analysis results, we decided to use only the 43 days that provided complete cross information for gross precipitation, stemflow and throughfall. Consequently, the total sample precipitation for this trial was reduced to 516.15 mm, with an average gross daily precipitation of 12.0 mm d$^{-1}$ and a daily range of 0.61 mm d$^{-1}$ to 40.26 mm d$^{-1}$.

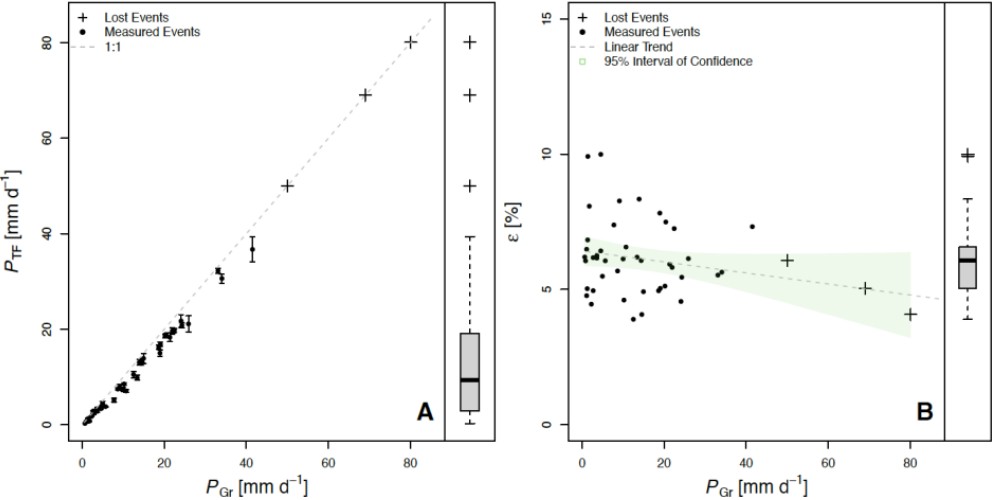

**Figure 2.** Linear relationship between throughfall and gross precipitation (**A**), and the distribution of the sampling error in percentage respect to gross precipitation (**B**) in a *Parkinsonia aculeata* stand at Palo Verde National Park, Costa Rica.

The selected trees to measure stemflow overcome the measured figures for tree diameter distribution of the whole plot (Table 1). Most of the trees in the subplots were multi-stem which restricted the possibility to set a more representative sample of single stem trees. The average of tree diameter among the subplots was 6.7 cm and ranged from 4.3 cm to 11.3 cm, while the selected trees had an average diameter of 8.8 cm with a range of 5.3 cm to 13.4 cm. Mean tree crown area for the selected trees was 10.6 m$^2$, with a range of 2.3 m$^2$ to 26.4 m$^2$.

After calculating the daily stemflow (mm m$^{-2}$) per each tree, we obtained the daily mean stemflow from the five selected trees. Then, we adjusted the daily mean stemflow

for each subplot and whole plot by multiplying the average stemflow in mm m$^{-2}$ to the estimated tree density of trees (trees m$^2$) for each plot (area of 100 m$^2$). In this case, the estimated tree densities were: whole plot (0.35), subplot 1 (0.09), subplot 2 (0.32) and subplot 3 (0.4). The results of this analysis are summarized in Table 2, which indicates that subplot 1 had the smallest total value of stemflow, only 0.68% of total gross precipitation, while subplots 2 and 3 resulted in comparable values of 3.24% and 3.99%, respectively, the average for the whole plot being 2.63%.

**Table 2.** Precipitation interception fluxes in a *Parkinsonia aculeata* L. stand at Palo Verde National Park, Costa Rica.

| Subplot | $P_{TF}$ (mm) | $P_{SF}$ (mm) | $P_{Net}$ (mm) | $P_I$ (mm) | $P_{TF}$ (%) | $P_{SF}$ (%) | $P_{Net}$ (%) | $P_I$ (%) |
|---|---|---|---|---|---|---|---|---|
| 1 | 471.8 | 3.5 | 475.2 | 40.9 | 91.40 | 0.68 | 92.07 | 7.93 |
| 2 | 450.2 | 16.7 | 466.9 | 49.3 | 87.21 | 3.24 | 90.45 | 9.55 |
| 3 | 452.0 | 20.6 | 472.6 | 43.6 | 87.57 | 3.99 | 91.56 | 8.44 |
| Whole Plot | 458.0 | 13.6 | 471.6 | 44.6 | 88.73 | 2.63 | 91.36 | 8.64 |

According to Table 2, subplot 1 has the lowest tree density and consequently had a higher value of throughfall than subplots 2 and 3. As it was observed in the field, all subplots, regardless of tree density, exhibited a complete canopy cover over each plot, explaining in part the small differences in throughfall rates among subplots.

For the regression analysis, we used daily gross precipitation as the independent variable and daily mean stemflow, throughfall and net precipitation as the dependent variables. First, we ran an ANOVA analysis comparing the $P_{TF}$ means and found that there were no statistical differences among the means of the subplots at $p < 0.05$. Therefore, we pooled all data from the three subplots (total of 30 gauges) to develop a generalized regression equation for the whole plot (Table 3, Figure 3). Then, we calculated the sampling error in percentage for the estimation of throughfall (Figure 1B). We found that for the gross precipitation events >10 mm d$^{-1}$ the sampling error for the estimation of mean throughfall was between 4% to 7%, while for the lowest daily gross precipitation events <10 mm d$^{-1}$ the sampling error ranged between 4% to 10%. Average throughfall sampling error for all gross precipitation events was 0.7 mm or 6.1%.

For the analysis of stemflow we observed that the relationship between gross precipitation and stemflow was nonlinear (Figure 3A), as expected. Therefore, we fitted a polynomial second degree equation with the intercept fixed to zero. A closer examination of the collected data revealed that there were 11 events under 2.8 mm d$^{-1}$ of gross precipitation that funneled insignificant values from 0.01 mm d$^{-1}$ to 0.04 mm d$^{-1}$. Hence, it can be suggested that stemflow in *P. aculeata* most likely will occur after 2.5 mm d$^{-1}$ of precipitation, and this value is larger to the estimated canopy storage capacity ($S_c$) determined with Equation (2) of 1.49 mm.

**Table 3.** Summary of linear regression analysis for water fluxes (y) against gross precipitation (x) in a *Parkinsonia aculeata* L. stand at Palo Verde National Park, Costa Rica.

| Water Fluxes | n | Regression Parameters | | | | Regression Coefficients | | |
|---|---|---|---|---|---|---|---|---|
| | | $R^2_{adj}$ | F | $p$ | SE | $\alpha$ | $\beta$ | $\gamma$ |
| $P_{TF}$ | 43 | 0.99 | 5336 | 0.000 | 0.83 | −0.5346 * | 0.9318 ** | |
| $P_{SF}$ | 43 | 0.89 | 240 | 0.000 | 0.16 | | 0.0014 ** | −0.0023 ** |
| $P_{Net}$ | 43 | 0.97 | 9675 | 0.000 | 0.97 | −0.7074 ** | 0.9475 ** | |

Note: n is the sample size, * is statistically significant at $p < 0.05$, ** is statistically significant at $p < 0.001$, SE is the standard error of the model.

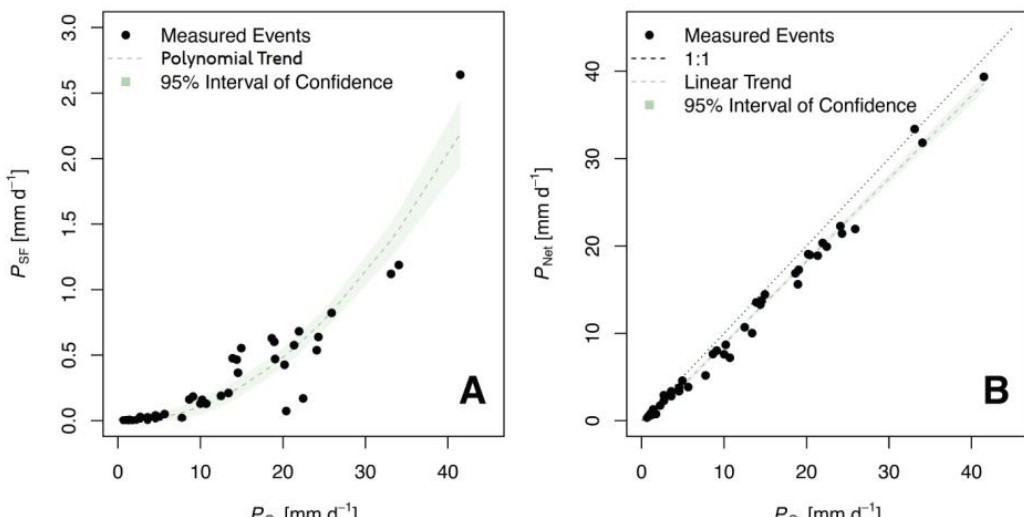

**Figure 3.** Relationships between daily stemflow (**A**) and net precipitation (**B**) against gross precipitation and the 95% confidence intervals for each of them in a *Parkinsonia aculeata* stand at Palo Verde National Park, Costa Rica.

Canopy storage capacity ($S_c$) determines the amount of water that the canopy can store before it saturates and starts dripping [34]. It depends on a wide number of characteristics such as tree shape, leaf surfaces, and superficial roughness of the bark [35,36]. The canopy storage effect has been widely studied in cloud forests [37]; however, its role in less complex ecosystems has not been fully studied yet. Llorens and Gallart [38] compiled a large set of $S_c$ across Europe showing the large variability among coniferous tree species (e.g., *Pinus silvestris*, *Picea abies*, *Pinus elliottii*) ranging from 0.1 mm to 3.1 mm. This parameter changes seasonally with forest phenology as it has been recorded for *Quercus brantii* in Iran, where $S_c$ changed from 0.56 mm to 1.56 mm between leafless and leafed periods, respectively [39]. In addition, the temperature plays an important role on the wettability of plant species [40], allowing some to store more water per unit surface at higher temperatures. This can explain why despite the relatively smooth surface of *P. aculeata*, the species is able to store a significant amount of water (1.49 mm) during the wet season.

The analyses of water fluxes in tree stands are common in the literature. There is a myriad of studies for large to medium sized tree species in different geographical settings. In the other hand, there is a notorious lack of studies for small trees to compare with the results of our study. However, there are two studies in the dry lands of Mexico for multi-stemmed small tree species comparable in height, diameter, branch distribution and crown shape with *P. aculeata*. Nevertheless, the used methodologies and sampling schemes in those two studies are not analogous to our study and consequently this discussion carries these limitations. The studies of Návar and Bryan [41] and Tamez-Ponce et al. [42] were in the state of Nuevo Leon, Mexico. The study of Návar and Bryan [41] sampled only 17 precipitation events (range of 2 mm d$^{-1}$ to 35 mm d$^{-1}$) and found for the small tree, *Diospyrus texana* Scheele (Ebenaceae), an average stemflow of 5.6% and throughfall of 67%. The study of Tamez-Ponce et al. [42] sampled 45 precipitation events (range of 14 to 56.5 mm) and focused on four small-medium sized trees, *Acacia farnesiana* (L.) Willd. (Fabaceae), *Condalia hookeri* M.C. Johnst. (Rhamnaceae), *Leucaena leucocephala* (Lam.) de Wit (Fabaceae) and *Casimiroa greggii* (S. Watson) F. Chiang (Rutaceae). For these species the study found the following throughfall estimates in percentages: 77%, 76%, 86% and 83%, respectively, while for stemflow the estimates oscillated from 1.12% for *C. hookeri* to 1.73% for *A. farnesiana*. Another study in Rockhampton, Australia, studied 19 tree species, with needle, broad and medium sizes leaves, in pure stands of three years of age [43]. We considered this study in this discussion because it provided information for trees in the early stages of growth, hence they are comparable with the height and diameter distribution for *P. aculeata*. The study in Australia sampled 50 precipitation events (range 0.6 mm d$^{-1}$ to 80 mm d$^{-1}$) and found an

average stemflow of 4.5% and an average throughfall of 30% (range 20.1% to 52.5%). It is within this context that we can conclude that *P. aculeata* has one of the highest throughfall rates (88.7%) and stemflow rates (2.6%) when compared to the small tree species. In general terms, *P. aculeata* shows comparable throughfall and stemflows with *L. leucocephala* and *C. greggii*.

The high throughfall values for *P. aculeata* (88.73%) could be explained by its canopy, which does not have a dense foliage like the other species, except for *L. leucocephala*. Even though the plots were well covered by the tree crowns during the rainy season, the canopy is more permeable to light and precipitation. This tree crown permeability is also related to two unique characteristics of *P. aculeata*: (a) the very thin flattened, waxy and ridged leaves rachis with tiny soft leaflets and (b) the soft bark of hanging twigs and small branches. All this makes the canopy low water retention increase the passage of water. On the other hand, the intermediate rate of stemflow of *P. aculeata* (2.63%) is the result of primary upward branches that efficiently funnel the water to the main trunk. Even though old primary branches have rough bark, the secondary branches likely have softer bark that contributes faster to stemflow [33,44]. Finally, the total interception loss of 8.64% seems to be one of the smallest rates among any size of tropical tree species.

The final linear regression equations in Table 3 are useful for estimating precipitation interception fluxes in *P. aculeata* and for hydrological modeling to study the impacts of the species on water balance, nutrient cycling, climate change analysis and active conservation management such as controlling the tree cover extension and density of this invasive species. These equations also can be applied in other geographical settings, as long as the range of gross precipitation and tree stand structural characteristics are comparable. In doing so, it must be remembered that the equation for stemflow estimates mm m$^2$, hence the data had to be adjusted to stand tree density.

## 4. Conclusions

As far as we know, this is the first study that generates valuable and precise information about precipitation interception and redistribution in *P. aculeata* worldwide. This invasive tree species showed to have one of the highest net precipitation and lowest precipitation interception among small trees. Nevertheless, an 8.64% loss of precipitation for canopy interception is a concern for the conservation of the Palo Verde Lagoon wetland, particularly during drought years and future climate warming scenarios. With this information, plus a digital elevation model of this wetland and good meteorological data, it is possible to model the effect of different *P. aculeata* cover scenarios to estimate all possible impacts on the water balance and hydrodynamics of this wetland.

**Author Contributions:** Conceptualization, methodology, funding acquisition, writing—original draft preparation, J.C.C.-A.; formal analysis, writing—review and editing, C.D.J.-R.; field-trial, O.A.-R.; mapping and forest inventory, J.C.S. All authors have read and agreed to the published version of the manuscript.

**Funding:** This research is part of the "Restoration project of Palo Verde lagoon wetland" promoted by the Organization for Tropical Studies with the sponsorship of CRUSA Foundation, AVINA Foundation, US Fish and Wildlife, and Vicerectoría de Investigación y Extensión of Instituto Tecnológico de Costa Rica with the project grant VIE 5402 1401 7301.

**Institutional Review Board Statement:** Not applicable.

**Informed Consent Statement:** Not applicable.

**Data Availability Statement:** The data set generated for this study can be found in ZENODO repository (https://doi.org/10.5281/zenodo.4820403, accessed on 27 May 2021).

**Acknowledgments:** Sincere thanks are given to Jorge Jiménez (Former OTS Director), Eugenio González (Former OTS Palo Verde Station Director), Mauricio Castillo, and José Guzman (OTS, GIS Lab-Palo Verde), for their help and the fruitful discussions concerning the hydrological restoration of the Palo Verde Lagoon.

**Conflicts of Interest:** The authors declare no conflict of interest.

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
