# Peer review of "Interception and Redistribution of Precipitation by Parkinsonia aculeata L.: Implications for Palo Verde National Park Wetlands, Costa Rica"

_water, doi:10.3390/w14030311_

Round 1

Reviewer 1 Report

Review of the manuscript: Interception and redistribution of precipitation by Parkinsonia aculeata L.: Implications for Palo Verde National Park wetlands, Costa Rica

General comments: The manuscript is well written in English and report the interception loss and redistribution of the precipitation of the Parkinsonia aculeata shrub in wetlands of Costa Rica, likewise outlines potential environmental influences in the water dynamics of this invasive species in the ecosystem. Also regarding the introduction and references, the interception loss and rainfall partitioning background are well documented. However, there are a number of internally conflicting statements that should be resolved before moving to publication.

Comments to authors:

Line 21: It is mentioned that the experiment was carried out in 2003. In this regard, a question immediately arises: What was the reason for not publishing those results back in 2003-2004?

Line 93: Regarding the previous point, it is mentioned that "understanding the current water balance dynamics of the wetlands", in this regard it is not possible to state that the study helps us understand the current situation of the water balance, since that wouldn’t be an accurate statement: your results from 18 years ago obviously cannot represent the current situation of the wetland or the structure of the vegetation. Do you believe that the results of 2003 are still valid today, even when the structure of the vegetation has changed?

Lines 107-109 and 175-178: It is mentioned that the climatic data of 2008-2017 were collated to describe the study area climate. It is correct to state that these climatic data represent the climatic situation of 2003, together with the current variation of climate due to climate change.

Line 128-129: Gross precipitation was measured only using one collector? No replications were made? If not, please explain more precisely the process for measuring gross rainfall. How did you estimate the mean gross rainfall with only one collector?

Lines 141-143: It is mentioned that the data collection took place daily, but it is not explained how you dealt with the problem of rain events that could have lasted more than 24 hours. Did you divide them as separate events or join them as a single event?  Please explain that process clearly within the methodology. Likewise, the number of rain events analyzed is not mentioned.

Lines 161-164: here is mentioned “individual gross precipitations event”, which is confused with the above mentioned of daily rainfall collection. Daily gross precipitation events vs individual rainfall events.

Line 164: Equation 1 (Eq 1) in which the net precipitation is defined is incorrect, unless you wanted to explain the interception loss.

 It says Pnet = PGr - PTF + PSF, which should be: Pnet = PTF + PSF

Line 173: What does Standard precipitation gauge mean? A pluviometer or a plastic funnel rain gauge?

Line 245: Says: “Tamez and Ponce et al”, should say: Tamez-Ponce et al.

Line 249: Says: “Tamez and Ponce et al”, should say: Tamez-Ponce et al.

Line 265: It says: P. aculeata (88,73%), it should say: P. aculeata (88.73%)

Line 307: According to table 3, 43 events were used to calculate the water flows in the ecosystem, does each point on the graph represent the mean value of 30 collectors? Likewise, the graph of the relationship between the interception loss and the gross precipitation was not shown, as well as its regression equation with its R2 in table 3, since Interception loss is the most important variable of the study.

Line 406: Its say “GraduaciónvMaestría”, should say “Graduación de Maestría”

Line 446: Please connect both family names of the authors i.e: Tamez-Ponce, Cantu-Silva, etc

Author Response

Response to Reviewer 1 Comments

In blue we copied the comments of the reviewer, in black our reply.

General comments: The manuscript is well written in English and report the interception loss and redistribution of the precipitation of the Parkinsonia aculeata shrub in wetlands of Costa Rica, likewise outlines potential environmental influences in the water dynamics of this invasive species in the ecosystem. Also regarding the introduction and references, the interception loss and rainfall partitioning background are well documented. However, there are a number of internally conflicting statements that should be resolved before moving to publication.

Comments to authors:

Line 21: It is mentioned that the experiment was carried out in 2003. In this regard, a question immediately arises: What was the reason for not publishing those results back in 2003-2004?

Reply:

The lead author (Julio Calvo-Alvarado) participated in different Costa Rica projects, producing large data sets, including long-term monitoring projects and short-term experiments. Also, in 2006 he took the lead of the Forestry School, and later, he became the President of the Tecnólogico de Costa Rica for eight years (TEC | Tecnológico de Costa Rica). This combination of research and administrative activities makes the previous analysis and publication of the data difficult. However, since 2019, the different databases have been checked and cleaned for their analysis, leading to various manuscripts.

Line 93: Regarding the previous point, it is mentioned that "understanding the current water balance dynamics of the wetlands", in this regard it is not possible to state that the study helps us understand the current situation of the water balance, since that wouldn’t be an accurate statement: your results from 18 years ago obviously cannot represent the current situation of the wetland or the structure of the vegetation. Do you believe that the results of 2003 are still valid today, even when the structure of the vegetation has changed?

Reply:

The authors agreed with the reviewer's point of view about the time difference between measurements and time of publication. However, the manuscript aims to generate relevant information about invasive species' impact that affects the wetland's hydrology. Despite the time-lapse since the measurements, P. aculeata is the second more critical invasive plant species after Typha on the marsh. Consequently, providing information about the redistribution of precipitation of a still present invasive plant cover is highly important.

Lines 107-109 and 175-178: It is mentioned that the climatic data of 2008-2017 were collated to describe the study area climate. It is correct to state that these climatic data represent the climatic situation of 2003, together with the current variation of climate due to climate change.

Reply:

We provided the 2008-2017 climatic data of the site to describe the climate on the site. This period was chosen because it is the longest and continuous record of meteorological conditions. The site has data from June 1999 to November 2007 with long data gaps. Both data sets were collected on the same spot but with different equipment. The first meteorological station was replaced due to various issues. We checked the first set of records, and between January 2000 and December 2005, the mean annual temperature was 27.6, and precipitation of 1023.55. There is a considerable difference in the annual rainfall. However, the year 2003 recorded 1400.4 mm of rain. This rain is closer to the 1645 mm recorded for 2008-2017. Consequently, we proposed first to update the lines 107-109 with the data from the period 2000-2005 as follows:

“… April). Data from the OTS automatic weather station indicates for the period 2000-2005 a mean annual precipitation of 1023.6±230.8 mm yr-1 and an average annual temperature of 27.6±0.3 °C, with the year 2003 registering 1400.4 mm yr-1 of rain.”

In the previous sentence update, we put the data set and site description in context. Therefore, the information in lines 175-178 provides the relevance of the experimental trial despite the time difference. In addition, we mentioned the second data set (2007-2020) as the current meteorological conditions, where the daily precipitation rates match the measured ones during the trial. Also, in line2 185-186, we clearly state the following:

“Thus, the sampling conducted in this study represents very well the long-term daily precipitation distribution of the site.”

Line 128-129: Gross precipitation was measured only using one collector? No replications were made? If not, please explain more precisely the process for measuring gross rainfall. How did you estimate the mean gross rainfall with only one collector?

Reply:

We used only one collector with no replicates because the size of the plot was not covering a large portion of land with significant heterogeneity in the spatial distribution of precipitation. Also, gross rainfall and throughfall were collected with the same device, reducing the experimental error during measurements. Finally, the closeness of the plot to the meteorological station allowed us to verify the recorded precipitation. Therefore, the manuscript never mentions mean gross precipitation because we measured it only with one collector and never stated the opposite.

Lines 141-143: It is mentioned that the data collection took place daily, but it is not explained how you dealt with the problem of rain events that could have lasted more than 24 hours. Did you divide them as separate events or join them as a single event?  Please explain that process clearly within the methodology. Likewise, the number of rain events analyzed is not mentioned.

Reply:

We measured the daily precipitation during the experiment, and those cases when the rainfall didn’t stop at 7:00 am were considered to pertain to the previous day during the analysis. Therefore, we consider daily precipitation (mm d-1) as an event in this manuscript. The authors think it is essential to add this detail in the methodology in line 143 as follows:

“The rainfall events that finished after or before the measuring time were assigned to the precipitation of the previous day. Consequently, this manuscript considers the daily precipitation (mm d-1) as an event and not multiple individual events within the same day.”

Lines 161-164: here is mentioned “individual gross precipitations event”, which is confused with the above mentioned of daily rainfall collection. Daily gross precipitation events vs individual rainfall events.

Reply:

With the addition of line 143 proposed in the previous reply, we clarify the meaning of rainfall events. However, to prevent the confusion that lines 161-164 may lead we proposed to modify those lines as follows:

“We calculated the sampling error (ε)  in percentage (%) according to Jiménez-Rodríguez and Calvo [33] to evaluate the precision of the sampling design. This estimation of error is based on the daily throughfall average of the whole plot with respect to daily gross precipitation.”

Line 164: Equation 1 (Eq 1) in which the net precipitation is defined is incorrect, unless you wanted to explain the interception loss.

 It says Pnet = PGr - PTF + PSF, which should be: Pnet = PTF + PSF

Reply:

Thanks to the reviewer. This error is a typo that we missed during the final editing. We will change it accordingly.

Line 173: What does Standard precipitation gauge mean? A pluviometer or a plastic funnel rain gauge?

Reply:

To clarify this point, we proposed to modify the line 173 as follows:

“The gross precipitation gauge collected …”

Line 245: Says: “Tamez and Ponce et al”, should say: Tamez-Ponce et al.

Reply:

Corrected accordingly.

Line 249: Says: “Tamez and Ponce et al”, should say: Tamez-Ponce et al.

Reply:

Corrected accordingly.

Line 265: It says: P. aculeata (88,73%), it should say: P. aculeata (88.73%)

Reply:

Corrected accordingly.

Line 307: According to table 3, 43 events were used to calculate the water flows in the ecosystem, does each point on the graph represent the mean value of 30 collectors?

Reply:

Each point corresponds to the mean value of the 30 collectors.

Likewise, the graph of the relationship between the interception loss and the gross precipitation was not shown, as well as its regression equation with its R2 in table 3, since Interception loss is the most important variable of the study.

Reply:

The authors agree with the reviewer’s statement that the interception loss is the most relevant variable of the study. However, the information related to the intercepted precipitation has not been left out in the manuscript. Table 2 provides the bulk (mm) and relative interception (%) values for the whole plot. The authors underline the importance of the precipitation redistribution in the manuscript and the usefulness of the net precipitation as a proxy to estimate the intercepted water by the canopy. Also, in Lines 247 to 259, we mention the importance of the canopy storage capacity as the critical variable that determines how much water the canopy can withhold.

Line 406: Its say “GraduaciónvMaestría”, should say “Graduación de Maestría”

Reply:

Corrected accordingly.

Line 446: Please connect both family names of the authors i.e: Tamez-Ponce, Cantu-Silva, etc

Reply:

Corrected accordingly.

Reviewer 2 Report

The manuscript presents the results from the interception process in Palo Verde National Park within the 3 subplots of P.aculeata forest with different tree densities. The results of gross and net precipitation via throughfall and stemflow measurements. Methods and experimental design were clearly and  correctly presented. Results were clearly presented and also discussed with other published papers.

I have some small remarks/recommentations:

  1. Are the Palo Verde National Park area influenced by advective foggy weather or not ?, because the fog occurrence could significantly influence the amount of interception loss.
  2. The stemflow calculations were made by using the canopy area of measured trees, did you try to calculate the stemflow values via the basal area of measured trees ? because for some trees this method has better results than canopy area calculations.
  3. Fig. 3A - dotted line is not linear trend
  4. Table 1 - tree density and basal area correct unit should be as tree.ha-1 and m2.ha-1

Author Response

Response to Reviewer 2 Comments

In blue we copied the comments of the reviewer, in black our reply.

The manuscript presents the results from the interception process in Palo Verde National Park within the 3 subplots of P. aculeata forest with different tree densities. The results of gross and net precipitation via throughfall and stemflow measurements. Methods and experimental design were clearly and correctly presented. Results were clearly presented and also discussed with other published papers.

I have some small remarks/recommendations:

1. Are the Palo Verde National Park area influenced by advective foggy weather or not?, because the fog occurrence could significantly influence the amount of interception loss.

Reply: Palo Verde National Park is not affected by advective foggy weather. The site is located close to the river mouth of Tempisque River in the Gulph of Nicoya. Despite its geographical location close to a large water body, the elevation difference of 8 m a.s.l. (Line: 103) does not allow this process to form.

2. The stemflow calculations were made by using the canopy area of measured trees, did you try to calculate the stemflow values via the basal area of measured trees? because for some trees this method has better results than canopy area calculations.

Reply: We did not try the methodology using basal area because the bushy characteristics of P. aculeata may induce an overestimation bias. Instead, using the tree's canopy area, we have a clear indication of the capture area of the sampled tree.

3. 3A - dotted line is not linear trend

Reply: We agree with the reviewer; this is a polynomial trend as stated in the methodology (Line: 152) and the results (Table 3). Therefore, we will update the image with the proper legend for Fig. 3A, adding Polynomial Trend instead of linear trend.

4. Table 1 - tree density and basal area correct unit should be as tree.ha-1 and m2.ha-1

Reply: Thanks for pointing out this typo. We will fix it on the manuscript.

Round 2

Reviewer 1 Report

The authors correctly answered each observation made to the document